# Neural Residual Diffusion Models for Deep Scalable Vision Generation

**Zhiyuan Ma[1], Liangliang Zhao[1,2], Biqing Qi[3], Bowen Zhou[1,3*]**
[1]Department of Electronic Engineering, Tsinghua University, Beijing, China
[2]Frontis.AI, Beijing, China
[3]Shanghai AI Laboratory, Shanghai, China
`{mzyth,zhoubowen}@tsinghua.edu.cn`

## Abstract

The most advanced diffusion models have recently adopted increasingly deep stacked networks (e.g., *U-Net* or *Transformer*) to promote the generative emergence capabilities of vision generation models similar to large language models (LLMs). However, progressively deeper stacked networks will intuitively cause numerical propagation errors and reduce noisy prediction capabilities on generative data, which hinders massively deep scalable training of vision generation models. In this paper, we first uncover the nature that neural networks being able to effectively perform generative denoising lies in the fact that the intrinsic residual unit has consistent dynamic property with the input signal's reverse diffusion process, thus supporting excellent generative abilities. Afterwards, we stand on the shoulders of two common types of deep stacked networks to propose a unified and massively scalable Neural Residual Diffusion Models framework (*Neural-RDM* for short), which is a simple yet meaningful change to the common architecture of deep generative networks by introducing a series of learnable gated residual parameters that conform to the generative dynamics. Experimental results on various generative tasks show that the proposed neural residual models obtain state-of-the-art scores on image's and video's generative benchmarks. Rigorous theoretical proofs and extensive experiments also demonstrate the advantages of this simple gated residual mechanism consistent with dynamic modeling in improving the fidelity and consistency of generated content and supporting large-scale scalable training.[1]

## 1 Introduction

Diffusion models (DMs) [1, 2, 3, 4] have emerged as a class of powerful generative models and have recently exhibited high quality samples in a wide variety of vision generation tasks such as image synthesis [5, 6, 7, 8, 9, 10, 11], video generation [12, 13, 14, 15, 16, 17, 18, 19] and 3D rendering and generation [20, 21, 22, 23, 24]. Relying on the advantage of iterative denoising and high-fidelity generation, DMs have gained enormous attention from the community and have been significantly improved in terms of sampling procedure [25, 26, 27, 28], conditional guidance [29, 30, 31, 32], likelihood maximization [33, 34, 35, 36] and generalization ability [37, 38, 39, 10] in previous efforts.

However, current diffusion models still face a scalability dilemma, which will play an important role in determining whether could support scalable deep generative training on large-scale vision data and give rise to emergent abilities [40] similar to large language models (LLMs) [41, 42]. Representatively, the recent emergence of Sora [43] has pushed the intelligent emergence capabilities of generative models to a climax by treating video models as world simulators. While unfortunately,

---

[*]Corresponding Author.
[1]Code is available at `https://github.com/ponyzym/Neural-RDM`.

Sora is still a closed-source system and the mechanism for the intelligence emergence is still not very clear, but the scalable architecture must be one of the most critical technologies, according to the latest investigation [44] on its reverse engineering.

To alleviate this dilemma and spark further research in the open source community beyond the realms of well established U-Net and Transformers, and enable DMs to be trained in new scalable deep generative architectures, we propose a unified and massively scalable *Residual-style Diffusion Models* framework (*Neural-RDM* for short) with a learnable gating residual mechanism, as shown in Figure 1.

The proposed *Neural-RDM* framework aims to unify the current mainstream residual-style generative architecture (e.g., *U-Net* or *Transformer*) and guide the emergence of brand new scalable network architectures with emergent capabilities. To achieve this goal, we first introduce a continuous-time neural ordinary differential equation (ODE) to prove that the generative denoising ability of the diffusion models is closely related to the residual-style network structure, which almost reveals the essential reason why any network rich in residual structure can denoise well: Residual-style neural units implicitly build an ordinary differential equation that can well fit the reverse denoising process through

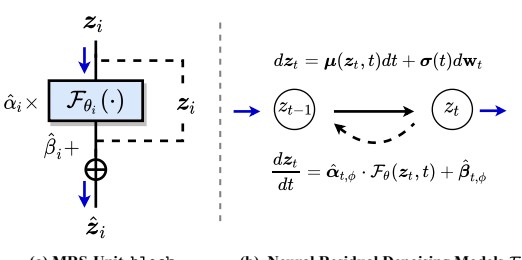

(a) MRS-Unit block$_i$      (b) Neural Residual Denoising Models $\mathcal{F}_\theta$

Figure 1: *Neural Residual-style Diffusion Models* framework with massively scalable gating-based **m**inimum **r**esidual **s**tacking **unit** (*mrs-unit*).

ever-deepening neural units, thus supporting excellent generative abilities. Further, we also show that the gating-residual mechanism plays an important role in adaptively correcting the errors of network propagation and approximating the mean and variance of data, which avoids the adverse factors of network deepening. On this basis, we further present the theoretical advantages of the *Neural-RDM* in terms of stability and score prediction sensitivity when stacking this residual units to a very long depth by introducing another residual-sensitivity ODE. From a dynamic perspective, it reveals that deep stacked networks have the challenge of gradually losing sensitivity as the network progressively deepens, and our proposed gating weights have advantages in reverse suppression and error control.

Our proposed framework has several theoretical and practical contributions:

**Unified residual denoising framework:** We unify the residual-style diffusion networks (e.g., *U-Net* and *Transformer*) by introducing a simple gating-residual mechanism and reveal the significance of the residual unit for effective denoising and generation from a brand new dynamics perspective.

**Theoretically deep scalability:** Thanks to the introduction of continuous-time ODE, we demonstrate that the dynamics equation expressed by deep residual networks possesses excellent dynamic consistency to the denoising probability flow ODE (PF-ODE) [45]. Based on this property, we achieve the simplest improvement to each mrs-unit by parameterizing a learnable mean-variance scheduler, which avoids to manually design and theoretically support massively deep scalable training.

**Adaptive stability maintenance and error sensitivity control:** When the mrs-units are infinitely stacked to express the dynamics of an overall network $\mathcal{F}_\theta$, the main technical difficulty is how to reduce the numerical errors caused by network propagation and ensure the stability of denoising. By introducing a sensitivity-related ODE in Sec. 2.3, we further demonstrate the theoretical advantages of the proposed gated residual networks in enabling stable denoising and effective sensitivity control. Qualitative and quantitative experimental results also consistently show their effectiveness.

## 2 Neural Residual Diffusion Models

We propose *Neural-RDM*, a simple yet meaningful change to the architecture of deep generative networks that facilitates effective denoising, dynamical isometry and enables the stable training of extremely deep networks. This framework is supported by three critical theories: **1)** *Gating-Residual* **ODE** (Sec. 2.1), which defines the dynamics of the **m**inimum **r**esidual **s**tacking **unit** (*mrs-unit* for short) that serves as the foundational denoising module, as shown in Figure 1 (a). Based on this gating-residual mechanism, we then introduce **2)** *Denoising-Dynamics* **ODE** (Sec. 2.2) to further stack the *mrs-units* to become a continuous-time deep score prediction network $\mathcal{F}_\theta$. Different from previous human-crafted *mean-variance* schedulers (e.g., variance exploding scheduler SMLD [46]

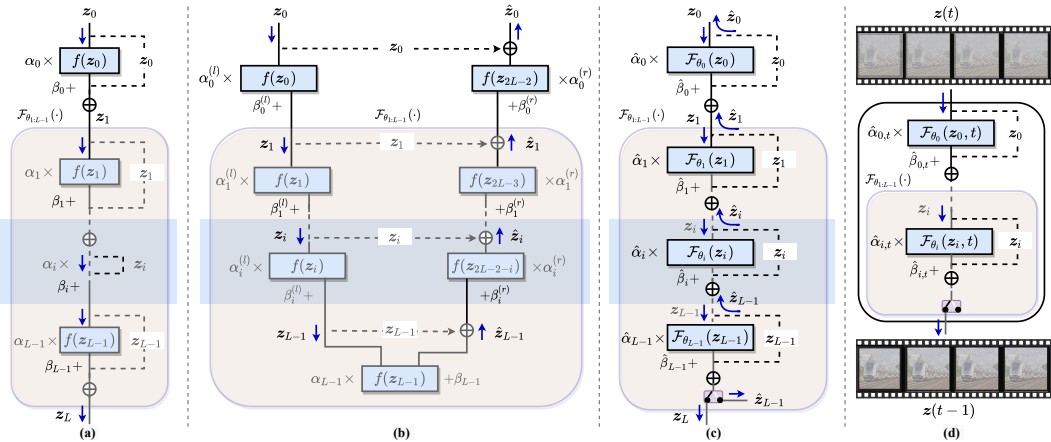

Figure 2: Overview. **(a)** Flow-shaped residual stacking networks. **(b)** U-shaped residual stacking networks. **(c)** Our proposed unified and massively scalable residual stacking architecture (i.e., *Neural-RDM*) with learnable gating-residual mechanism. **(d)** Residual denoising process via *Neural-RDM*.

and variance preserving scheduler DDPM [2]), which may cause concerns about instability in denoising efficiency and quality, we introduce a parametric method to implicitly learn the mean and variance distribution, which lowers the threshold of manual design and enhances the generalization ability of models. Last but not least, to maintain the stability of the deep stacked networks and verify the sensitivity of each residual unit $\mathcal{F}_{\theta_i}(\cdot)$ to the network $\mathcal{F}_{\theta}$, we stand on the shoulders of the *adjoint sensitivity* method [47, 48] to propose **3) *Residual-Sensitivity* ODE** (Sec. 2.3), which means the sensitivity-related dynamics of each latent state $z_i$ from $\mathcal{F}_{\theta_i}(\cdot)$ to the deep network $\mathcal{F}_{\theta}$. Through rigorous derivation, we prove that the parameterized gating weights have a positive inhibitory effect on sensitivity decaying as network deepening. We will elaborate on them below.

## 2.1 Gating-Residual Mechanism

Let $\mathcal{F}_{\theta_i}$ represents the minimum residual unit $\mathtt{block}_i$ (Figure 1 (a)), $f(\cdot)$ denotes any feature mapper wrapped by $\mathcal{F}_{\theta_i}$. Instead of propagating the signal $z$ through each of vanilla neural transformation $\hat{z} = f_\theta(z)$ directly, we introduce a gating-based residual connection for the signal $z$, which relys on the two learnable gating weights $\hat{\alpha}$ and $\hat{\beta}$ to modulate the non-trivial transformation $\mathcal{F}_{\theta_i}(z_i)$ as,

$$\hat{z}_i = z_i + \hat{\alpha}_i \cdot \mathcal{F}_{\theta_i}(z_i) + \hat{\beta}_i. \tag{1}$$

For a deep neural network $\mathcal{F}_{\theta}(\cdot)$ with depth $L$, consider two common residual stacking fashions: Flow-shaped Stacking (**FS**) [49, 50] and U-shaped Stacking (**US**) [51, 52]. For the flow-based deep stacking networks as shown in Figure 2 (a), each residual unit $f(\cdot)$ accepts the output $z_i$ of the previous *mrs-unit* as input, and obtains a new hidden state $z_{i+1}$ through gating-residual connection,

$$\hat{z}_i = z_{i+1} = z_i + [\alpha_i \cdot f_{\theta_i}(z_i) + \beta_i]. \tag{2}$$

Note that Eq. 2 is a refined form of Eq. 1 in the case of flow-shaped stacking. In contrast, for the U-shaped deep stacking networks as in Figure 2 (b), each minimum residual unit contains two symmetrical branches, where the left branch receives the output $z_i$ of the previous *mrs-unit*'s left branch as input (called the *read-in* branch), and the right branch performs the critical nonlinear residual transformation for readout (called the *read-out* branch), which can be formally described as:

$$\hat{z}_i = \underbrace{\alpha_i^{(l)} \cdot f_{\theta_i^{(l)}}(z_i) + \beta_i^{(l)}}_{\text{read-in branch}} \hookrightarrow \underbrace{z_i + \alpha_i^{(r)} \cdot f_{\theta_i^{(r)}}(z_{2L-2-i}) + \beta_i^{(r)}}_{\text{read-out branch}} = z_i + \hat{\alpha}_i \cdot \mathcal{F}_{\theta_i}(z_i) + \hat{\beta}_i. \tag{3}$$

Here Eq. 3 is a refined form of Eq. 1 in the case of U-shaped stacking, $\hat{\alpha}_i$ and $\hat{\beta}_i$ collectively denotes the gating weights from the left and right branches, $\mathcal{F}_{\theta_i}$ is the $i$-th minimum residual unit of the U-shaped networks, and "$\hookrightarrow$" denotes the skip-connection for "$z_{i+1} \to z_{2L-2-i}$", which is computed recursively via $\mathcal{F}_{\theta_{i+1:L-1}}$. To enable the networks to be infinitely stacked, we introduce a continuous-time *Gating-Residual* ordinary differential equation (ODE) to express the neural dynamics

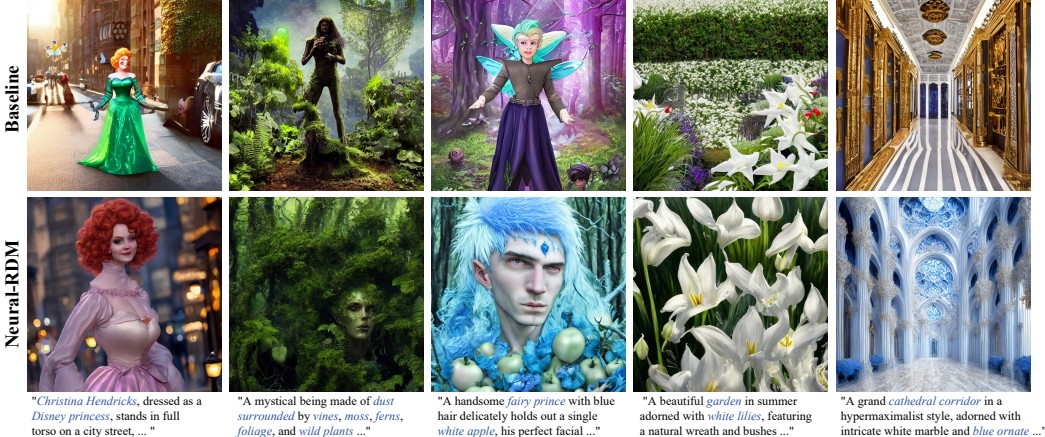

Figure 3: Compared with the latest baseline (SDXL-1.0 [7]), the samples produced by Neural-RDM (trained on JourneyDB [53]) exhibit exceptional quality, particularly in terms of fidelity and consistency in the details of the subjects in adhering to the provided textual prompts.

of these two types of deep stacking networks ($\delta = \frac{1}{L}$, $L \to \infty$ denotes the number of the *mrs-units*),

$$\frac{\boldsymbol{z}_{i+\delta} - \boldsymbol{z}_i}{\delta} = \hat{\boldsymbol{z}}_i - \boldsymbol{z}_i = \hat{\alpha}_i \cdot \mathcal{F}_{\theta_i}(\boldsymbol{z}_i) + \hat{\beta}_i \implies \frac{d\boldsymbol{z}_t}{dt} = \hat{\alpha}_\phi \cdot \mathcal{F}_{\theta_t}(\boldsymbol{z}_t) + \hat{\beta}_\phi, \qquad (4)$$

where $\phi$ represents the gating weights, which can be independently trained or fine-tuned without considering the parameters $\theta$ of the feature mapping network $\mathcal{F}_\theta(\cdot)$ itself.

## 2.2 Denoising Dynamics Parameterization

The above-mentioned gating-residual mechanism is utilized to modulate mainstream deep stacking networks and unify them into a residual-style massively scalable generative framework, as shown in Figure 2 (c). Next, we further explore the essential relationship between residual neural networks and score-based generative denoising models from a dynamic perspective.

First, inspired by the theory of continuous-time diffusion models [45, 54], the forward add-noising process can be expressed as a dynamic process with stochastic differential equation (SDE) as,

$$d\boldsymbol{z}_t = \boldsymbol{\mu}(\boldsymbol{z}_t, t)dt + \boldsymbol{\sigma}(t)d\mathbf{w}_t \implies \frac{d\boldsymbol{z}_t}{dt} = \boldsymbol{\mu}(\boldsymbol{z}_t, t) + \boldsymbol{\sigma}(t) \cdot \epsilon_t, \epsilon_t \in \mathcal{N}(\mathbf{0}, \mathbf{I}), \qquad (5)$$

which describes a data perturbation process controlled by a *mean-variance* scheduler composed of $\boldsymbol{\mu}(\boldsymbol{z}_t, t)$ and $\boldsymbol{\sigma}(t)$ respectively, $\mathbf{w}_t$ denotes the standard Brownian motion. Compared with the forward process, the core of the diffusion model is to utilize a deep neural network (as deep and large as possible) for score-based reverse prediction [46, 55]. A remarkable property of this SDE is the existence of a reverse ODE (also dubbed as the *Probability Flow* (PF) ODE by [45]), which retain the same marginal probability densities as the SDE (See Appendix. A.2 for detailed proof) and could effectively guide the dynamics of the reverse denoising, it can be formally described as,

$$\frac{d\boldsymbol{z}_t}{dt} = \boldsymbol{\mu}(\boldsymbol{z}_t, t) - \frac{1}{2}\boldsymbol{\sigma}(t)^2 \cdot \left[\nabla_z \log p_t(\boldsymbol{z}_t)\right] = \hat{\boldsymbol{\alpha}}_{t,\phi} \cdot \mathcal{F}_\theta(\boldsymbol{z}_t, t) + \hat{\boldsymbol{\beta}}_{t,\phi}, \qquad (6)$$

where $\nabla_z \log p_t(\boldsymbol{z}_t)$ denotes the gradient of the log-likelihood of $p_t(\boldsymbol{z}_t)$, which can be estimated by a score matching network $\mathcal{F}_\theta(\boldsymbol{z}_t, t)$. Here we re-parameterize the PF-ODE by utilizing gated weights to replace the manually designed mean-variance scheduler, in which $\hat{\boldsymbol{\alpha}}_{t,\phi}$ and $\hat{\boldsymbol{\beta}}_{t,\phi}$ denotes the time-dependent dynamics parameters, which is respectively parameterized to represent $-\frac{1}{2}\boldsymbol{\sigma}(t)^2$ and $\boldsymbol{\mu}(\boldsymbol{z}_t, t)$ by our proposed gating-residual mechanism. Note that $\mathcal{F}_\theta(\cdot)$ is a score estimation network composed of infinite *mrs-units* $\texttt{block}_i$ (i.e., $\mathcal{F}_{\theta_i}$), which enables massively scalable generative training on large-scale vision data, but also presents the challenge of numerical propagation errors.

## 2.3 Residual Sensitivity Control

To control the numerical errors in back-propagation and achieve steadily and massively scalable training, we stand on the shoulders of the *adjoint sensitivity* method [47, 48] to introduce another

| Architecture | Method | Scalability | Class-to-Image | | | Text-to-Image | | |
|---|---|---|---|---|---|---|---|---|
| | | | FID↓ | sFID↓ | IS↑ | FID↓ | sFID↓ | IS↑ |
| GAN | BigGAN-deep [56] | ✗ | 6.95 | 7.36 | 171.4 | - | - | - |
| | StyleGAN-XL [57] | ✗ | 2.30 | 4.02 | 265.12 | - | - | - |
| U-shaped | ADM [58] | ✔ | 10.94 | 6.02 | 100.98 | - | - | - |
| | ADM-U | ✔ | 7.49 | 5.13 | 127.49 | - | - | - |
| | ADM-G | ✔ | 4.59 | 5.25 | 186.70 | - | - | - |
| | LDM-8 [30] | ✔ | 15.51 | - | 79.03 | 16.64 | 11.32 | 64.50 |
| | LDM-8-G | ✔ | 7.76 | - | 209.52 | 9.35 | 10.02 | 125.73 |
| | LDM-4 | ✔ | 10.56 | - | 103.49 | 12.37 | 11.58 | 94.65 |
| | LDM-4-G | ✔ | 3.60 | - | 247.67 | 3.78 | 5.89 | 182.53 |
| F-shaped | DiT-XL/2 [59] | ✔ | 9.62 | 6.85 | 121.50 | 8.53 | 5.47 | 144.26 |
| | DiT-XL/2-G | ✔ | 2.27 | 4.60 | 278.24 | 3.53 | 5.48 | 175.63 |
| | Latte-XL [60] | ✔ | 2.35 | 5.17 | 224.75 | 2.74 | 5.35 | 195.03 |
| Unified | **Neural-RDM-U (Ours)** | ✔✔ | 3.47 | 5.08 | 256.55 | **2.25** | **4.36** | **235.35** |
| | **Neural-RDM-F (Ours)** | ✔✔ | **2.12** | **3.75** | **295.32** | 2.46 | 5.65 | 206.32 |

Table 1: The main results for image generation on ImageNet [61] (Class-to-Image) and Jour-neyDB [53] (Text-to-Image) with $256 \times 256$ image resolution. We highlight the best value in blue, and the second-best value in green. The *Scalability* column indicates the scaling capability of the parameter scale and architecture.

*Residual-Sensitivity* ODE, which is utilized to evaluate the sensitivity of each residual-state $z_t$ of the *mrs-unit* $\mathcal{F}_{\theta_i}$ to the total loss $\mathcal{L}$ derived by score estimation network $\mathcal{F}_\theta(\cdot)$ (the sensitivity is denoted as $s_t = \frac{d\mathcal{L}}{dz_t}$, $\delta$ denotes an infinitesimal time interval) and can be formally described by the chain rule,

$$s_t = \frac{d\mathcal{L}}{dz_t} = \frac{d\mathcal{L}}{dz_{t+\delta}} \cdot \frac{dz_{t+\delta}}{dz_t} = s_{t+\delta} \cdot \frac{dz_{t+\delta}}{dz_t}. \tag{7}$$

On the basis of Eq. 7, we next continue to discuss the dynamic equation of sensitivity changing with time $t$. First, considering the trivial transformation $f_\theta(\cdot)$ without gating-residual mechanism,

$$dz_{t+\delta} = dz_t + \int_t^{t+\delta} f_\theta(z_t, t)dt. \tag{8}$$

We can rewrite Eq. 7 based on Eq. 8 as:

$$s_t = s_{t+\delta} + s_{t+\delta} \cdot \frac{\partial}{\partial z_t}(\int_t^{t+\delta} f_\theta(z_t, t)dt). \tag{9}$$

The *Residual-Sensitivity* ODE under vanilla situation then can be derived,

$$\frac{ds_t}{dt} = \lim_{\delta \to 0^+} \frac{s_{t+\delta} - s_t}{\delta} = \lim_{\delta \to 0^+} \frac{-s_{t+\delta} \cdot \frac{\partial}{\partial z_t}(\int_t^{t+\delta} f_\theta(z_t)dt)}{\delta} = -s_t \cdot \frac{\partial f_\theta(z_t, t)}{\partial z_t}. \tag{10}$$

According to the derived residual-sensitivity equation in Eq. 10, we further use the Euler solver to obtain the sensitivity $s_{t_0}$ of the starting state $z_{t_0}$ to network $\mathcal{F}_\theta(\cdot)$ as,

$$s_{t_0} = s_{t_L} + \int_{t_L}^{t_0} \frac{ds_t}{dt}dt = s_{t_L} - \int_{t_L}^{t_0} s_t \cdot \frac{\partial f_\theta(z_t, t)}{\partial z_t}dt. \tag{11}$$

Due to the non-negativity of the integral and the gradient $\frac{\partial f_\theta(z_t, t)}{\partial z_t}$ not equals to 0, we can obtain a gradually decaying sensitivity sequence: $s_{t_L} > s_{t_{L-1}} > \cdots > s_{t_0}$. Similarly, when defining parameter-sensitivity $s_\theta = \frac{d\mathcal{L}}{d\theta}$, the same decaying results for $s_{\theta_0}$ can be obtained:

$$s_{\theta_0} = s_{\theta_L} + \int_{t_L}^{t_0} \frac{ds_\theta}{dt}dt = s_{\theta_L} - \int_{t_L}^{t_0} s_\theta \cdot \frac{\partial f_\theta(z_t, t)}{\partial \theta}dt. \tag{12}$$

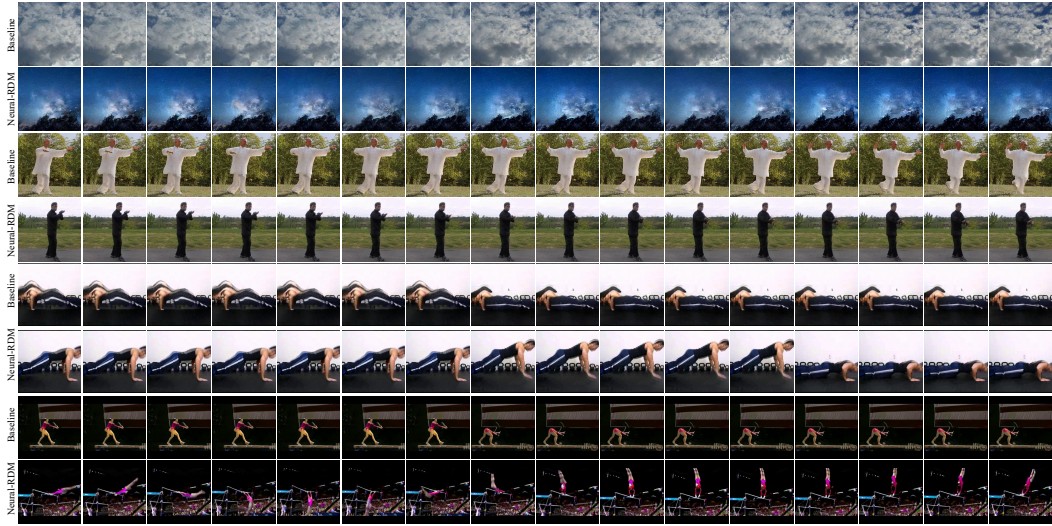

Figure 4: Compared with the latest baseline (Latte-XL [60]), the sample videos from SkyTime-lapse [62], Taichi-HD[63] and UCF101 [64] all exhibit better frame quality, temporal consistency and coherence.

To alleviate this problem, and enable stable training in massively deep scalable architecture, we introduce the following non-trivial solution with gating-residual transformation,

$$d\hat{\boldsymbol{z}}_{t+\delta} = d\hat{\boldsymbol{z}}_t + \int_t^{t+\delta} \left[\alpha_{t,\phi} \cdot f_\theta(\hat{\boldsymbol{z}}_t) + \beta_{t,\phi}\right] dt. \tag{13}$$

Substitute Eq. 13 into Eq. 7 to obtain the corrected sensitivity $\hat{\boldsymbol{s}}_t = \frac{d\mathcal{L}}{d\hat{\boldsymbol{z}}_t}$ as:

$$\hat{\boldsymbol{s}}_t = \hat{\boldsymbol{s}}_{t+\delta} + \hat{\boldsymbol{s}}_{t+\delta} \cdot \frac{\partial}{\partial \hat{\boldsymbol{z}}_t} \left(\int_t^{t+\delta} \left[\alpha_{t,\phi} \cdot f_\theta(\hat{\boldsymbol{z}}_t) + \beta_{t,\phi}\right] dt\right). \tag{14}$$

The non-trivial *Residual-Sensitivity* ODE can be derived as,

$$\frac{d\hat{\boldsymbol{s}}_t}{dt} = \lim_{\delta \to 0^+} \frac{\hat{\boldsymbol{s}}_{t+\delta} - \hat{\boldsymbol{s}}_t}{\delta} = -(\alpha_{t,\phi} \cdot \hat{\boldsymbol{s}}_t) \cdot \frac{\partial f_\theta(\hat{\boldsymbol{z}}_t, t)}{\partial \hat{\boldsymbol{z}}_t} - (\beta_{t,\phi} \cdot \hat{\boldsymbol{s}}_t). \tag{15}$$

Through the Euler solver, we can also obtain the sensitivity $\hat{\boldsymbol{s}}_{t_0}$ of the starting state adjusted by the gating-residual weights,

$$\hat{\boldsymbol{s}}_{t_0} = \hat{\boldsymbol{s}}_{t_L} + \int_{t_L}^{t_0} \frac{d\hat{\boldsymbol{s}}_t}{dt} dt = \hat{\boldsymbol{s}}_{t_L} - \int_{t_L}^{t_0} \left[(\alpha_{t,\phi} \cdot \hat{\boldsymbol{s}}_t) \cdot \frac{\partial f_\theta(\hat{\boldsymbol{z}}_t, t)}{\partial \hat{\boldsymbol{z}}_t} + (\beta_{t,\phi} \cdot \hat{\boldsymbol{s}}_t)\right] dt. \tag{16}$$

Where $\alpha_{t,\phi}$ and $\beta_{t,\phi}$ adaptively modulate and update the sensitivity of each *mrs-unit* to the final loss, which supports being trained through minimizing $\mathcal{L}_s = ||\mathcal{F}_\theta(\boldsymbol{z}_t, t) - \nabla_z \log p_t(\boldsymbol{z}_t)||_2^2 + \gamma \cdot \sum_L ||\alpha_{t,\phi} \cdot \frac{\partial f_\theta(\hat{\boldsymbol{z}}_t, t)}{\partial \hat{\boldsymbol{z}}_t} - \beta_{t,\phi}||_2^2$ in full-parameter training or model fine-tuning fashions.

## 3 Experiments

We present the main experimental settings in Sec. 3.1. To evaluate the generative performance of Neural-RDM, we compare it with state-of-the-art conditional/unconditional diffusion models for image synthesis and video generation in Sec. 3.2 and Sec. 3.3 respectively. We also visualize and analyze the effects of the proposed gated residuals and illustrate their advantages in enabling deep scalable training, which are presented in Sec. 3.4 and Sec. 3.5.

### 3.1 Experimental Settings

**Datasets.** For image synthesis tasks, we train and evaluate the **Class-to-Image** generation models on the ImageNet [61] dataset and train and evaluate the **Text-to-Image** generation models on the

| Method | Scalability | Frame Evaluation | | None-to-Video | | Class-to-Video |
| | | FID↓ | IS↑ | SkyTimelapse (FVD↓) | Taichi-HD (FVD↓) | UCF-101 (FVD↓) |
|---|---|---|---|---|---|---|
| MoCoGAN [71] | ✗ | 23.97 | 10.09 | 206.6 | - | 2886.9 |
| MoCoGAN-HD [72] | ✗ | 7.12 | 23.39 | 164.1 | 128.1 | 1729.6 |
| DIGAN [73] | ✗ | 19.10 | 23.16 | 83.11 | 156.7 | 1630.2 |
| StyleGAN-V [70] | ✗ | 9.45 | 23.94 | 79.52 | - | 1431.0 |
| MoStGAN-V [74] | ✗ | - | - | 65.30 | - | 1380.3 |
| PVDM [75] | ✔ | 29.76 | 60.55 | 75.48 | 540.2 | 1141.9 |
| LVDM [12] | ✔ | - | - | 95.20 | 99.0 | **372.0** |
| VideoGPT [76] | ✔ | 22.70 | 12.61 | 222.7 | - | 2880.6 |
| Latte-XL [60] | ✔ | 5.02 | 68.53 | 59.82 | 159.60 | 477.97 |
| **Neural-RDM (Ours)** | ✔✔ | 3.35 | 85.97 | **39.89** | **91.22** | **461.03** |

Table 2: The main results for video generation on the SkyTimelapse [62], Taichi-HD [63] and UCF-101 [64] with $256 \times 256$ resolution of each frame. We highlight the best value in  blue , and the second-best value in  green .

MSCOCO [65] and JourneyDB [53] datasets. All images are resized to $256 \times 256$ resolution for training. For video generation tasks, we follow the previous works [12, 60] to train **None-to-Video** (*i.e.,* unconditional video generation) models on the SkyTimelapse [62] and Taichi [63] datasets, and train **Class-to-Video** models on the UCF-101 [64] dataset. Moreover, we follow previous works [12, 60] to sample 16-frame video clips from these video datasets and then resize all frames to $256 \times 256$ resolution for training and evaluation.

**Implementation details.** We implement our Neural-RDMs into Neural-RDM-U (U-shaped) and Neural-RDM-F (Flow-shaped) two versions on top of the current state-of-the-art diffusion models LDM [30] and Latte [60] for image generation, and further employ the Neural-RDM-F version for video generation. Specifically, we first load the corresponding pre-trained models and initialize gating parameters $\{\alpha = 1, \beta = 0\}$ of each layer, then perform full-parameter fine-tuning to implicitly learn the distribution of the data for acting as a parameterized mean-variance scheduler. During the training process, we adopt an explicit supervision strategy to enhance the sensitivity correction capabilities of $\alpha$ and $\beta$ for deep scalable training, where the explicitly supervised hyper-parameter $\gamma$ is set to 0.35. Eventually, we utilize the AdamW optimizer with a constant learning rate of $5 \times 10^4$ for all models and exploit an exponential moving average (EMA) strategy to obtain and report all results.

**Evaluation metrics.** Following the previous baselines [30, 58, 59, 60], we adopt Fréchet Inception Distance (FID) [66], sFID [67] and Inception Score (IS) [68] to evaluate the image generation quality and the video frame quality (except for sFID). Furthermore, we utilize a Fréchet Video Distance (FVD) [69] metric similar with FID to evaluate the unconditional and conditional video generation quality. Among these metrics, FVD is closer to human subjective judgment and thus better reflects the visual quality of the generated video content. Adhering to the evaluation guidelines proposed by StyleGAN-V [70], we calculate the FVD scores by analyzing 2048 generated video clips with each clip consists of 16 frames.

**Baselines.** We compare the proposed method with the recent state-of-the-art baselines, and categorize them into three groups: 1) **GAN-based.** BigGAN-deep [56] and StyleGAN-XL [57] for image task, MoCoGAN [71], MoCoGAN-HD [72], DIGAN [73], StyleGAN-V [70] and MoStGAN-V [74] for video task. 2) **U-shaped.** ADM [58] and LDM [30] for image task, PVDM [75] and LVDM [12] for video task. 3) **F-shaped.** DiT-XL/2 [59] and Latte-XL [60] for image task, VideoGPT [76] and Latte-XL [60] (with temporal attention learning) for video task.

## 3.2 Experiments on Image Synthesis with Deep Scalable Spatial Learning

For a more objective comparison, we maintain approximately the same model size to perform class-conditional and text-conditional image generation experiments, which are shown in Table 1. From Table 1, it can be observed that our Neural-RDMs have obtained state-of-the-art results. Specifically,

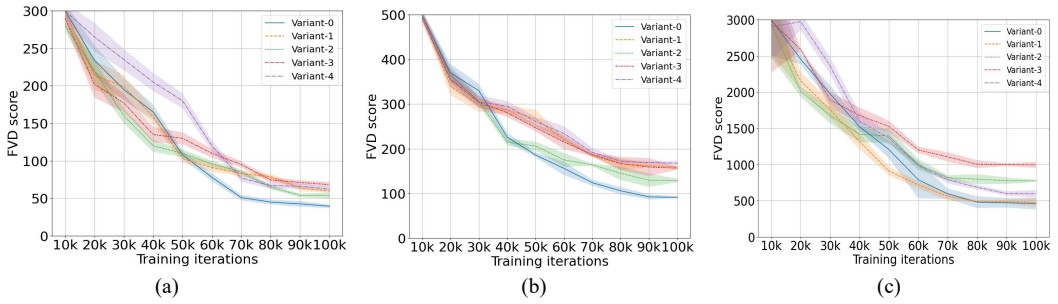

(a)                                          (b)                                          (c)

Figure 6: **(a)**, **(b)**, and **(c)** respectively illustrate the performance of the five residual structures variant models across the SkyTimelapsee [62], Taichi-HD[63], and UCF-101 [64].

the flow-based version (i.e., Neural-RDM-F) consistently outperforms all class-to-image baselines in all three image's generative benchmarks and meanwhile obtains relatively suboptimal results on another text-to-image evaluations. It is worth noting that another Neural-RDM-U version have made up for this shortcoming and achieved optimal results, which may benefit from the more powerful semantic guidance abilities of the cross-attention layer built into U-Net. To more clearly present the actual effects of the gated residuals, we further perform qualitative comparative experiments, which are shown in Figure 3. Compared with the latest baseline (SDXL-1.0 [7]), we can observe that the samples produced by Neural-RDM exhibit exceptional quality, particularly in terms of fidelity and consistency in the details of the subjects in adhering to the provided textual prompts, which consistently demonstrates the effectiveness of our proposed approach in deep scalable spatial learning.

### 3.3 Experiments on Video Generation with Deep Scalable Temporal Learning

To further explore the effectiveness and specific contributions of proposed gating-residual mechanism in temporal learning, we continue to perform the video generation evaluations, which are shown in Table 2. From Table 2, we find that our model (flow-shaped version) basically achieves the best results (except for the second-best results in **class-to-video** evaluation). Specifically, compare with Latte-XL [60], Neural-RDM respectively achieves an improvement of 33.3% and 42.8% in FVD scores on Sky-Timelapse and Taichi-HD datasets, which hints the powerful potential of flow-based deep residual networks in promoting generative emergent capabilities of video models. Furthermore, we exhibit a number of visual comparison results of the 16-frames video produced by Neural-RDM and baseline (Latte-XL [60]), as shown in Figure 4. We can observe that some generated frames from the baseline partially exhibits poor quality and temporal inconsistency. Compare with the baseline, Neural-RDM maintains tem-

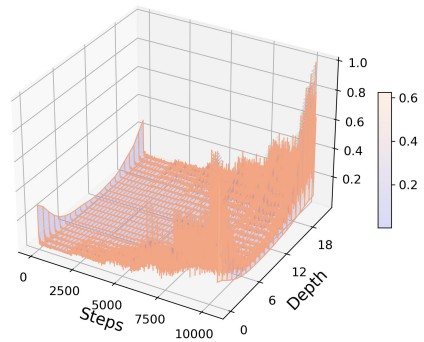

Figure 5: The sensitivity of $\alpha$ and $\beta$ at different depths of the residual denoising network during the training process.

poral coherence and consistency, resulting in smoother and more dynamic video frames, which further reflects the effectiveness of proposed method in both quantitative and qualitative evaluations.

### 3.4 The Analyses of Gating Residual Sensitivity

To better illustrate the advantage of the gated residuals and understand the positive suppression effect for sensitivity attenuation as network deepening, we visualize the normalized sensitivity at different depths of our Neural-RDM during the training process, as shown in Figure 5. From Figure 5, we can observe that $\alpha$ and $\beta$ can adaptively modulate the sensitivity of each *mrs-unit* to correct the denoising process as network deepening, which is consistent with Eq. 16. Moreover, we can also observe that at the beginning of training, the sensitivity scores are relatively low. As training advances, $\alpha$ and $\beta$ are supervised to correct the sensitivity until obtaining relatively higher sensitivity scores.

## 3.5 Comparison Experiments of Gating Residual Variants and Deep Scalability

To explore the actual effects of different residual settings in deep training, we first perform the comparison experiments on 5 different residual variants: **1)** *Variant-0 (Ours)*: $z_{i+1} = z_i + \alpha f(z_i) + \beta$; **2)** *Variant-1 (AdaLN [77])*: $z_{i+1} = z_i + f(\alpha z_i + \beta)$; **3)** *Variant-2*: $z_{i+1} = \alpha z_i + f(z_i) + \beta$; **4)** *Variant-3 (ResNet [78])*: $z_{i+1} = z_i + f(z_i)$; **5)** *Variant-4 (ReZero [79])*: $z_{i+1} = z_i + \alpha f(z_i)$.

We utilize Latte-XL as backbone to train each variant from scratch and then evaluate their performance for video generation. As depicted in Figure 6, as the number of training steps increases, almost all variants can converge effectively, but only *Variant-0* (Our approach) achieves the best FVD scores. We speculate that it may be because this post-processing gating-residual setting maintains excellent dynamic consistency with the reverse denoising process, thus achieving better performance.

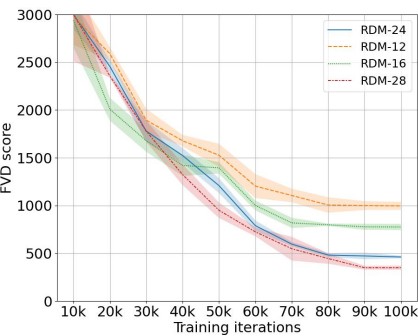

Moreover, we further perform the deeep scalability experiments, which are shown in Figure 7. We can observe that as the depth of residual units increases, the performance of the model can be further improved, which illustrates the positive correlation between model performance and the depth of residual units and further highlights the deep scalability advantage of our Neural-RDM.

Figure 7: The performance of Neural-RDM with different network depths on the UCF-101 dataset [64].

## 4 Related Work

**Deep Residual Networks.** Most common deep residual networks can be divided into two types of architectures: flow-shaped stacking (FS) and u-shaped stacking (US) architectures. As a milestone of flow-based deep residual networks, ResNet [78] has led the research of visual understanding tasks [80]. In fact, the pratices [81, 82] and theories [83, 84, 85] that introducing the highway connections [86, 87] have been studied for a long time and have demonstrated excellent advantages in dealing with vanishing/exploding gradients and numerical propagation errors in deep stacked networks. Different from ResNet, U-Net [51] is a leader of u-shaped networks and almost dominated diffusion-based generative models [2, 30]. Though achieving remarkable success, both types of CNN-based models still face concerns about training efficiency. Recent years, Transformer [49] and ViT [50] have emerged as new state-of-the-art backbones in computer vision and multimodal [88, 89, 90, 91] and have also gained prominence in various diffusion models. Among them, DiT [59] and U-ViT [52] are two representative works by respectively adopting flow-shaped and u-shaped residual stacking fashions, which have enabled many studies on deep generative models [60, 92, 93, 94, 95]. In this work, we unify the above two types of residual stacking architectures from a dynamic perspective and propose a unified and deep scalable neural residual framework with a same gating-residual ODE.

**Diffusion Models.** Recent years has witnessed the remarkable success of diffusion models [2, 3, 4], due to their impressive generative capabilities. Previous efforts mainly focus on sampling procedure [25, 26, 27, 28], conditional guidance [31, 32, 96, 97, 98], likelihood maximization [33, 34, 35, 36] and generalization ability [37, 99, 39, 10] and have gained enormous attention. Recently, a major research topic on diffusion models is scalability. DiT [59] is one of the most representative models by exploiting a scalable Transformer to train latent diffusion models for image generation. Latte [60] stands on the shoulders of DiT to further perform temporal learning for video generation. However, both Latte and DiT adopt the residual structure of Transformer by default and utilize S-AdaLN to incorporate guidance information, they generally lack: 1) attention to the residual structure and 2) study the dynamic nature of the deep generative models, and 3) ignore the error propagation issues from deeper networks and therefore are still limited by the bottleneck of massively scalable training.

Overall, we practically unify u-shaped and flow-shaped stacking networks and to propose a unified and deep scalable neural residual diffusion model framework. Moreover, we theoretically parameterize the previous human-designed mean-variance scheduler and demonstrate excellent dynamics consistency.

## 5 Conclusion

In this paper, we have presented Neural-RDM, a simple yet meaningful change to the architecture of deep generative networks that facilitates effective denoising, dynamical isometry and enables the stable training of extremely deep networks. Further, we have explored the nature of two common types of neural networks that enable effective denoising estimation. On this basis, we introduce a parametric method to replace previous human-designed mean-variance schedulers into a series of learnable gating-residual weights. Experimental results on various generative tasks show that Neural-RDM obtains the best results, and extensive experiments also demonstrate the advantages in improving the fidelity, consistency of generated content and supporting large-scale scalable training.

## Acknowledgments and Disclosure of Funding

This work is supported by the National Science and Technology Major Project (2023ZD0121403), National Natural Science Foundation of China (No. 62406161), China Postdoctoral Science Foundation (No. 2023M741950), and the Postdoctoral Fellowship Program of CPSF (No. GZB20230347).

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

# A  Appendix

## A.1  Theoretical Interpretations

In this section, we provide mathematical intuitions for our Neural-RDMs.

**Continuous-time Residual Networks.**  For a deep neural network $\mathcal{F}_\theta(\cdot)$ with depth $L$, let $\mathcal{F}_{\theta_i}$ represents the minimum residual unit $\texttt{block}_i$ (Figure 1 (a)). Instead of propagating the signal $z$ through vanilla neural transformation $\hat{z} = f_\theta(z)$, we introduce a gating-based skip-connection for the signal $z$, which relys on the gating weights $\hat{\alpha}$ and $\hat{\beta}$ to modulate the non-trivial transformation $\mathcal{F}_\theta(z)$ as,

$$\hat{z} = z + \hat{\alpha} \cdot \mathcal{F}_\theta(z) + \hat{\beta}. \tag{17}$$

In the case of continuous time, this dynamic equation describing the change process of the signal $z$ is called the ***gating-residual* ODE**:

$$\frac{d\boldsymbol{z}_t}{dt} = \hat{\alpha}_\phi \cdot \mathcal{F}_{\theta_t}(\boldsymbol{z}_t) + \hat{\beta}_\phi, \tag{18}$$

**Diffusion Probability Models.**  The diffusion probability models are modeled as: 1) a deterministic forward noising process $q(\mathbf{x}_t|\mathbf{x}_{t-1}) = \mathcal{N}(\mathbf{x}_t; \sqrt{\alpha_t}\mathbf{x}_{t-1}, (1-\alpha_t)\mathbf{I})$ from the original image $\mathbf{x}_0$ to a pure-Gaussian distribution $\mathbf{x}_T \sim \mathcal{N}(0, \mathbf{I})$, which can be formulated in an accumulated form:

$$\mathbf{x}_t = \sqrt{\bar{\alpha}_t}\mathbf{x}_0 + \sqrt{1-\bar{\alpha}_t}\boldsymbol{\epsilon}, \quad \boldsymbol{\epsilon} \sim \mathcal{N}(0, \mathbf{I}) \tag{19}$$

2) a iteratively predictable reverse denoising process $p_\theta(\mathbf{x}_{t-1}|\mathbf{x}_t) = \mathcal{N}(\mathbf{x}_{t-1}; \boldsymbol{\mu}_\theta(\mathbf{x}_t, t), \boldsymbol{\Sigma}_\theta(\mathbf{x}_t, t))$, which can be trained in a simplied denoising objective $\mathcal{L}_{\text{simple}}$ by merging $\boldsymbol{\mu}_\theta$ and $\boldsymbol{\Sigma}_\theta$ into predicting noise $\boldsymbol{\epsilon}_\theta$,

$$\mathcal{L}_{\text{simple}} = E_{\mathbf{x}_0, t, \boldsymbol{\epsilon} \sim \mathcal{N}(0, \mathbf{I})}[||\boldsymbol{\epsilon} - \boldsymbol{\epsilon}_\theta(\mathbf{x}_t, t)||_2^2] \tag{20}$$

where $t \sim \mathcal{U}[1, T]$ is time parameters, $\mathcal{U}(\cdot)$ denotes uniform distribution. Moreover, in Stable Diffusion [30], the image $\mathbf{x}_t$ is compressed into a latent variable $\mathbf{z}_t$ by encoder $\mathcal{E}$ for more efficient training, i.e., $\mathbf{z}_t = \mathcal{E}(\mathbf{x}_t)$, thus this preliminary objective is usually defined as making $\boldsymbol{\epsilon}_\theta(\mathbf{z}_t, t)$ as close to $\boldsymbol{\epsilon} \sim \mathcal{N}(0, \mathbf{I})$ as possible.

**Reverse Denoising ODE.**  A remarkable property of the SDE (Eq. 19) is the existence of a reverse ODE (also dubbed as the *Probability Flow* (PF) ODE by [45]), which retains the same marginal probability densities as the SDE (See Appendix A.2 for detailed proof) and could effectively guide the dynamics of the reverse denoising, it can be formally described as,

$$\frac{d\boldsymbol{z}_t}{dt} = \boldsymbol{\mu}(\boldsymbol{z}_t, t) - \frac{1}{2}\boldsymbol{\sigma}(t)^2 \cdot \left[\nabla_z \log p_t(\boldsymbol{z}_t)\right] = \hat{\boldsymbol{\alpha}}_{t,\phi} \cdot \mathcal{F}_\theta(\boldsymbol{z}_t, t) + \hat{\boldsymbol{\beta}}_{t,\phi}, \tag{21}$$

where $\nabla_z \log p_t(\boldsymbol{z}_t)$ denotes the gradient of the log-likelihood of $p_t(\boldsymbol{z}_t)$, which can be estimated by a score matching network $\mathcal{F}_\theta(\boldsymbol{z}_t, t)$.

**Dynamics Consistency.**  Refer to Eq. 18 and Eq. 21, we define this dynamic consistency as: For any time-dependent signal $\boldsymbol{z}_t$, the different dynamics systems describe it with the same motion path (or the same change rate of data distribution). Note that in Eq. 21, we achieve this by using a re-parameterized approach.

**Latent Space Projection.**  The latent space projection is proposed by [30] to compress the input images $\mathbf{x}_0$ into a perceptual high-dimensional space to obtain $\mathbf{z}_0$ by leveraging a pretrained VQ-VAE model [100]. The VQ-VAE is also used by our Neural-RDM, it consists of an encoder $\mathcal{E}$ and a decoder $\mathcal{G}$. The mathematical definition is: Given an input image $x \in \mathbb{R}^{H \times W \times 3}$, the VQ-VAE first compress the image $x$ into a latent variable $\hat{z}$ by encoder $\mathcal{E}$, i.e., $\hat{z} = \mathcal{E}(x)$ and $\hat{z} \in \mathbb{R}^{h \times w \times d}$, where $h$ and $w$ respectively denote scaled height and width (scaled factor $f = H/h = W/w = 8$), and $d$ is the dimensionality of the compressed latent variable. After going through the diffusion step described in Eq. 5 and Eq. 6, the latent variable $\hat{z}$ is updated and finally reconstructed into $\hat{x}$ by decoder $\mathcal{G}$,

$$\hat{x} = \mathcal{G}_\pi(\text{LDM}_{\mathcal{F}_\theta(\cdot)}(\mathcal{E}_\pi(x))), \tag{22}$$

where $\text{LDM}(\cdot)$ represents the latent diffusion models (including Unet-based or Transformer-based), $\theta$ denotes the parameters of LDM, and $\pi$ denotes the parameters of the VQVAE that are frozen to train our Neural-RDM models.

## A.2 Additional Proofs

Motivated by [101], we follow [45] to give a proof: A remarkable property of the SDE (Eq. 5) is the existence of a reverse ODE (PF-ODE [45]), which retain the same marginal probability densities as the SDE. We consider the SDE in Eq. 5, which possesses the following form:

$$d\boldsymbol{z}_t = \boldsymbol{\mu}(\boldsymbol{z}_t, t)dt + \boldsymbol{\sigma}(\boldsymbol{z}_t, t)d\mathbf{w}_t, \tag{23}$$

where $\boldsymbol{\mu}(\cdot, t) : R^d \to R^d$ and $\boldsymbol{\sigma}(\cdot, t) : R^d \to R^{d \times d}$. The marginal probability density $p_t(\boldsymbol{z}_t)$ evolves according to Kolmogorov's forward equation [102]

$$
\frac{\partial p_t(\boldsymbol{z})}{\partial t} = -\sum_{i=1}^d \frac{\partial}{\partial z_i}[\mu_i(\boldsymbol{z}_t, t)p_t(\boldsymbol{z}_t)] + \frac{1}{2}\sum_{i=1}^d\sum_{j=1}^d \frac{\partial^2}{\partial z_i \partial z_j}\Big[\sum_{k=1}^d \sigma_{ik}(\boldsymbol{z}_t, t)\sigma_{jk}(\boldsymbol{z}_t, t)p_t(\boldsymbol{z}_t)\Big].
$$

$$
= -\sum_{i=1}^d \frac{\partial}{\partial z_i}[\mu_i(\boldsymbol{z}_t, t)p_t(\boldsymbol{z}_t)] + \frac{1}{2}\sum_{i=1}^d \frac{\partial}{\partial z_i}\Big[\sum_{j=1}^d \frac{\partial}{\partial z_j}\Big[\sum_{k=1}^d \sigma_{ik}(\boldsymbol{z}_t, t)\sigma_{jk}(\boldsymbol{z}_t, t)p_t(\boldsymbol{z}_t)\Big]\Big]. \tag{24}
$$

Since the sub-part of Eq. 24 can be written in the following form:

$$
\sum_{j=1}^d \frac{\partial}{\partial z_j}\Big[\sum_{k=1}^d \sigma_{ik}(\boldsymbol{z}_t, t)\sigma_{jk}(\boldsymbol{z}_t, t)p_t(\boldsymbol{z}_t)\Big]
$$

$$
= \sum_{j=1}^d \frac{\partial}{\partial z_j}\Big[\sum_{k=1}^d \sigma_{ik}(\boldsymbol{z}_t, t)\sigma_{jk}(\boldsymbol{z}_t, t)\Big]p_t(\boldsymbol{z}_t) + \sum_{j=1}^d\sum_{k=1}^d \sigma_{ik}(\boldsymbol{z}_t, t)\sigma_{jk}(\boldsymbol{z}_t, t)p_t(\boldsymbol{z}_t)\frac{\partial}{\partial z_j}\log p_t(\boldsymbol{z}_t)
$$

$$
= p_t(\boldsymbol{z}_t)\nabla \cdot [\boldsymbol{\sigma}(\boldsymbol{z}_t, t)\boldsymbol{\sigma}^\top(\boldsymbol{z}_t, t)] + p_t(\boldsymbol{z}_t)\boldsymbol{\sigma}(\boldsymbol{z}_t, t)\boldsymbol{\sigma}^\top(\boldsymbol{z}_t, t)\nabla_{\boldsymbol{z}_t}\log p_t(\boldsymbol{z}_t). \tag{25}
$$

Thus we can obtain:

$$
\frac{\partial p_t(\boldsymbol{z}_t)}{\partial t} = -\sum_{i=1}^d \frac{\partial}{\partial z_i}[\mu_i(\boldsymbol{z}_t, t)p_t(\boldsymbol{z}_t)] + \frac{1}{2}\sum_{i=1}^d \frac{\partial}{\partial z_i}\Big[\sum_{j=1}^d \frac{\partial}{\partial z_j}\Big[\sum_{k=1}^d \sigma_{ik}(\boldsymbol{z}_t, t)\sigma_{jk}(\boldsymbol{z}_t, t)p_t(\boldsymbol{z}_t)\Big]\Big]
$$

$$
= -\sum_{i=1}^d \frac{\partial}{\partial z_i}[\mu_i(\boldsymbol{z}_t, t)p_t(\boldsymbol{z}_t)]
$$

$$
+ \frac{1}{2}\sum_{i=1}^d \frac{\partial}{\partial z_i}\Big[p_t(\boldsymbol{z}_t)\nabla \cdot [\boldsymbol{\sigma}(\boldsymbol{z}_t, t)\boldsymbol{\sigma}^\top(\boldsymbol{z}_t, t)] + p_t(\boldsymbol{z}_t)\boldsymbol{\sigma}(\boldsymbol{z}_t, t)\boldsymbol{\sigma}^\top(\boldsymbol{z}_t, t)\nabla_{\boldsymbol{z}_t}\log p_t(\boldsymbol{z}_t)\Big]
$$

$$
= -\sum_{i=1}^d \frac{\partial}{\partial z_i}\Big\{\mu_i(\boldsymbol{z}_t, t)p_t(\boldsymbol{z}_t)
$$

$$
- \frac{1}{2}\Big[\nabla \cdot [\boldsymbol{\sigma}(\boldsymbol{z}_t, t)\boldsymbol{\sigma}^\top(\boldsymbol{z}_t, t)] + \boldsymbol{\sigma}(\boldsymbol{z}_t, t)\boldsymbol{\sigma}^\top(\boldsymbol{z}_t, t)\nabla_{\boldsymbol{z}_t}\log p_t(\boldsymbol{z}_t)\Big]p_t(\boldsymbol{z}_t)\Big\}
$$

$$
= -\sum_{i=1}^d \frac{\partial}{\partial z_i}[\tilde{\mu}_i(\boldsymbol{z}_t, t)p_t(\boldsymbol{z}_t)], \tag{26}
$$

where we define $\tilde{\boldsymbol{\mu}}_i(\cdot)$ as:

$$
\tilde{\boldsymbol{\mu}}(\boldsymbol{z}_t, t) := \boldsymbol{\mu}(\boldsymbol{z}_t, t) - \frac{1}{2}\nabla \cdot [\boldsymbol{\sigma}(\boldsymbol{z}_t, t)\boldsymbol{\sigma}^\top(\boldsymbol{z}_t, t)] - \frac{1}{2}\boldsymbol{\sigma}(\boldsymbol{z}_t, t)\boldsymbol{\sigma}^\top(\boldsymbol{z}_t, t)\nabla_{\boldsymbol{z}_t}\log p_t(\boldsymbol{z}_t). \tag{27}
$$

Combining Eq. 26 and Eq. 27, we can conclude that Equation Eq. 26 still describes a Kolmogorov's forward process but with $\tilde{\boldsymbol{\sigma}}(\boldsymbol{z}_t, t) := \mathbf{0}$ as:

$$
d\boldsymbol{z}_t = \tilde{\boldsymbol{\mu}}(\boldsymbol{z}_t, t)dt + \tilde{\boldsymbol{\sigma}}(\boldsymbol{z}_t, t)d\mathbf{w}_t, \quad \tilde{\boldsymbol{\sigma}}(\boldsymbol{z}_t, t) = \mathbf{0}. \tag{28}
$$

Which proves that it is actually an ODE after reverse transformation $\tilde{\boldsymbol{\mu}}(\cdot)$:

$$
d\boldsymbol{z}_t = \tilde{\boldsymbol{\mu}}(\boldsymbol{z}_t, t)dt, \tag{29}
$$

which is essentially the same with our ***Denoising-Diffusion*-ODE** given by Eq. 6. Therefore, we demonstrate the existence of the reverse ODE and the practicality of parameterizing the mean-variance scheduler from the reverse ODE.

## A.3   More Generated Images

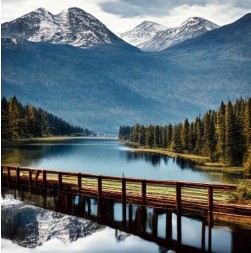 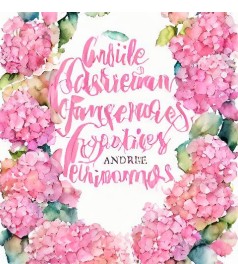 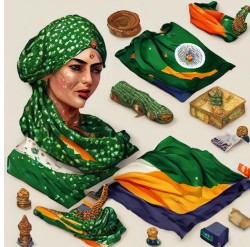 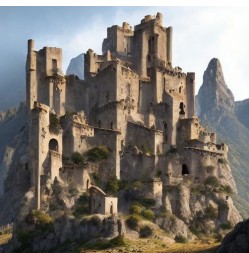

"A *landscape* featuring a *wooden bridge* over a serene *lake* with a majestic *mountain* in the background."

"A cartoon-style *watercolor* cover illustration featuring *vibrant pink* and *cream hydrangeas* in a Disney-inspired setting, complemented by typography."

"A speckled *headscarf*, a swamp adder, and a *flag of India* depicted in an isometric illustration."

"A *castle* situated in the *mountains* with an array of very high thin *towers* adorned with numerous arrowslits."

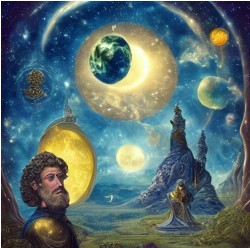 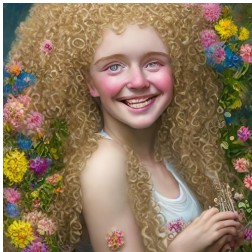 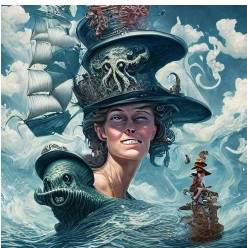 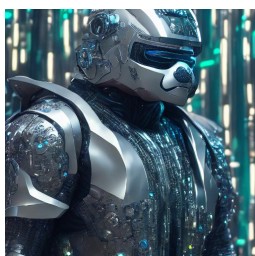

"The *King* of Pentacles stands in a regal pose, surrounded by earthly riches, while a *mysterious UFO* hovers above, adding an element of otherworldly intrigue."

"A hyper-realistic portrait of a 17-year-old *English girl* with mismatched eyes, *blonde curly hair* adorned with flowers, holding a flute, radiating pure joy."

"*woman captain* wearing a *cocked hat* stands on a ship, gazing in awe and fear as a huge Cthulhu emerges from the water."

"A *cyberpunk man* with silver skin wearing a helmet featuring a *large glass visor,* holding a rocket launcher in a futuristic setting, depicted with hyper-realistic..."

Figure 8: The samples produced by Neural-RDM (trained on JourneyDB [53]) .

## A.4   Limitations

**Limitation Discussion.**   Although significant improvements have been achieved, Neural-RDM still has some limitations, the most important of which is that the gated residual mechanism only inhibits rather than completely avoids the sensitivity decrease and numerical propagation errors caused by the deepening of the network. If we want to completely avoid it, we may have to give up stacking-based deep network architectures, but that will lead to a significant reduction in performance. Therefore, our method chooses to continue to deepen the stacking of the network and suppress error propagation as much as possible in the trade-off between the two.

## A.5   Social Impact

**Potential Social Implications.**   We believe that Neural-RDM will bring new thinking about deep network architectures to the generative community, and hopefully promote the generative emergence capabilities of vision generation models in the open source community. In addition, we hope that more researchers can follow the powerful capabilities of residual denoising to build brand new scalable network architectures beyond the realms of well established U-Net and Transformers.

