# OpenReview forum: "Neural Residual Diffusion Models for Deep Scalable Vision Generation"
_NeurIPS.cc/2024/Conference — NeurIPS 2024 poster_

### Official Review · Reviewer_6rnE · 2024-07-11

**Soundness:** 3
**Presentation:** 2
**Contribution:** 3
**Rating:** 6
**Confidence:** 4

**Summary:**

The paper describes the gradual change in the state z_t of a diffusion-based neural network as an ODE, and links the flow-architecture and UNet-architecture to the parameters of the ODE. Then, it describes a way to better train this network based on the dynamics of this ODE, and proposes an alternative loss function. It then performs experiments on image and video generation with this loss function, and several variants of the architecture.

**Strengths:**

The paper does a great job of explaining how the dynamics of z_t through the network can be unified across the UNet and Flow architectures. It then derives a loss function based on the ODE of those dynamics. The derivation itself is involved and highly useful. The paper also performs several experiments on multiple modalities to validate the proposed hypothesis of the new loss function / perspective. In particular, there are experiments in both image and video generation, each being quite strenuous in terms of academic effort and computational requirements.

---
Update: I increased my score after the authors' response.

**Weaknesses:**

Some parts of the paper are not clearly written. In Figure 2, the caption does not sufficiently explain the figure. In particular, (a) and (c) look the same, the difference needs to be elaborated on (corresponding to the main text). What dotted lines mean, what coloured lines mean, etc. has to be explained, in Figures 1 and 2.

Sections 2.2 and 2.3 need to be written more clearly. In particular, the dimensions of the involved variables need to be mentioned. It is unclear whether the network is used to completely or partially denoise the input.

Most image-based experiments seem to be finetuned experiments, which are dependent on the biases of the original network used. Section 3.5 contains experiments on models trained from scratch, which is great. It is unclear though why Variant 0 performs so much better than Variant 4, since they only have a scalar "beta" difference.

**Questions:**

Is the final loss function as stated in lines 162-163? In which case, does the loss apply to each layer separately? Do you sum the losses per layer?

How was the value of gamma as 0.35 decided?

It is unclear whether the network is used to completely or partially denoise the input. It seems as though it is assumed that the network is completely denoising from noise to data (instead of transforming partially noised to slightly less noised). Because the dynamics of F mention L -> infinite. If, however, F is assumed to be the full sampling procedure of multiple denoising steps, then Figure 2 doesn't make sense. In brief, it is unclear how the full dynamics are connected with each forward pass of the network. This is especially confusing from the perspective of the UNet.

The difference between Variant 0 and Variant 4 seems to be just a scalar beta value. Why does this 1 parameter affect performance so much?

**Limitations:**

The paper states where the model works better than prior works and where it doesn't.

---

> ### Author Rebuttal · Authors · 2024-08-06
>
> **Q1: Explanation of some details in Fig.1 & 2**\
> **R1:** Thanks. We will explain these details below:
> 1) In Fig.2, (a), (b) and (c) respectively represent **Flow-shaped Networks** (e.g., Transformer), **U-shaped Networks** (e.g., U-Net) and **Unified Stacking Network (our Neural-RDM)**. As depicted in line 98-118, (a) and (b) are both special cases of (c), i.e., $f_{\theta_i}$ in (a) corresponds to $F_{\theta_i}$in (c), whereas $f_{\theta_i^{(l)}}$ and $f_{\theta_i^{(r)}}$ in (b) jointly correspond to $F_{\theta_i}$ in (c).  Note that (c) covers both data streams mentioned earlier with a unified dynamics formula, which is a significant difference from (a).
> 2) The dotted lines in Fig.1 & 2 indicate the residual connections.
> 3) The coloured arrows (i.e., blue arrows) denote the data streams of the stacked networks.
>
> We will carefully supplement these details in the final version.
>
> **Q2: Dimensions of the variables**\
> **R2:** We first want to clarify that this paper is actually a foundational work applicable to a variety of DMs' deep-scalable training, e.g., image, video and others, thus we do not specifically emphasize the dimensions of the variables, but they are actually easy to specify, e.g., $\boldsymbol{z}\in\mathbb{R}^{B \times F \times C \times H \times W}$($B$ is bach size, $F$ is video frames, $C$ is channels, $H$ and $W$ denote height and width respectively), $\alpha\in\mathbb{R}^{L \times D}$ and $\beta\in\mathbb{R}^{L  \times D}$ ($L$ is the number of network layers, $D$ is dimension of the hidden layer) in video tasks. Note that the image tasks are slightly different from the videos in frame dimensions. If necessary, we'll add them in the final version.
>
> **Q3: Whether to denoise partially or completely**\
> **R3:** We suspect this may be a serious misunderstanding, and we want to clarify that although the Neural-RDM network ($F_{\theta}$) is composed of a series of mrs-unit $F_{\theta_i}$, but it is still performed for single-step denoising, i.e., $\boldsymbol{z}(t)\rightarrow\boldsymbol{z}(t-1)$ for time t, as illustrated in Fig.2(d). Note that this point is completely consistent with the previous baseline LDM, DiT and Latte. Moreover, we want to further highlight: we introduce continuous-time **_Residual-Sensitivity ODE_** into the neural networks of depth L, **NOT** into the diffusion process (e.g., $\boldsymbol{z}(0)\rightarrow\boldsymbol{z}(T)$). Moreover, for U-shaped stacking networks, we want to specify $\mathcal{F} _ {\theta _ {i}}$ stands for both $f_{\theta_i^{(l)}}$ and $f_{\theta_i^{(r)}}$ in the i-th residual unit, i.e., the skip-connection $\boldsymbol{z} _ {i+1}\rightarrow\boldsymbol{z}_{2L-2-i}$ in UNet or U-ViT. The goal is to build a unified **_Denoising-Dynamics ODE_** under consistent dynamics formula, so as to facilitate the improvement and optimization of all stacked networks (including U-shaped and Flow-shaped) for better deep scalable training.
> We will make these points clearer in the final revision and hope the above clarifications can help understand our work better.
>
> **Q4: Explanation for the difference in performance between variant 0 and variant 4**\
> **R4:** We understand the reviewer's concern, but that's actually unnecessary. Firstly, we want to clarify that the $\hat{\beta} _ {t,\phi}$ is not a simple scalar, but rather a series of learnable time-dependent parameters that are used to fit the mean-related scheduler (i.e., $\mu(z_t,t)$) in our **_Denoising-Dynamics ODE_**, thus the adjustment of $\hat{\beta} _ {t,\phi}$ directly impacts the denoising performance (i.e. the FVD score). More broadly, the $\hat{\alpha} _ {t,\phi}$ and $\hat{\beta} _ {t,\phi}$ respectively parameterized the variance- and mean-related scheduler $-\frac{1}{2}\sigma(t)^2$ and $\mu(\boldsymbol{z} _ t,t)$, which replaces human design in previous works as,
>
> $\frac{d\boldsymbol{z} _ t}{dt} = \mu(\boldsymbol{z} _ t,t)-\frac{1}{2}\sigma(t)^2\cdot\Big[\nabla _ {z} \log p_t(\boldsymbol{z}_t)\Big] = \hat{\alpha} _ {t,\phi}\cdot\mathcal{F} _ \theta(\boldsymbol{z} _ t,t) + \hat{\beta} _ {t,\phi}.$
>
> Whereas variant 4 contains only a scaling tensor $\hat{\alpha}_{t,\phi}$, making it difficult to fit the **_Denoising-Dynamics ODE_**, thus resulting in worse results than variant 0.
>
> **Q5: Details of the Loss Function**\
> **R5:** Yes, the final loss function with sensitivity control is as stated in lines 162-163, which can be expressed as follows,
>
> $\mathcal{L} _ s = ||\mathcal{F} _ \theta(\boldsymbol{z} _ {t}, t) - \nabla_z \log p _ t(\boldsymbol{z} _ t)||^2_2 + \gamma \cdot \sum _ {L}|| \hat{\alpha}_ {t,\phi} \cdot\frac{\partial f _ \theta(\hat{\boldsymbol{z}} _ {t},t)}{\partial\hat{\boldsymbol{z}} _ {t}}-\hat{\beta}_{t,\phi}||^2_2.$
>
> The total loss is obtained by summing the loss per layer, with sensitivity of each mrs-unit layer $F_ {\theta_i}$ adaptively modulated and updated by $\hat{\alpha} _ {t,\phi}$ and $\hat{\beta} _ {t,\phi}$.
>
> **Q6: Determination of hyper-parameter $\gamma$**\
> **R6:** The value of the $\gamma$ is determined via hyper-parameter search experiments. As shown in the table below, we conducted extensive experiments with different values of $\gamma$ on C2I & T2I tasks, benchmarking on ImageNet and JourneyDB datasets respectively. Based on the observation from experimental results, we select $\gamma=0.35$ as the optimal value.
> | $\gamma$ |  | 0.00 | 0.20 | 0.30 | 0.35* | 0.40 | 0.60 | 0.80 | 1.00 |
> | --- | --- | --- | --- | --- | --- | --- | --- | --- | --- |
> | C2I (IS$\uparrow$)  | Neural-RDM-U | 79.03 | 140.24 | 241.61 | **256.55** | 228.00 | 173.52 | 152.41 | 139.92 |
> |  | Neural-RDM-F | 224.75 | 247.07 | 273.09 | **295.32** | 266.91 | 263.06 | 247.71 | 235.79 |
> | T2I (IS$\uparrow$)  | Neural-RDM-U | 64.50 | 182.52 | 269.98 | **235.35** | 208.04 | 192.67 | 189.27 | 146.35 |
> |  | Neural-RDM-F | 195.03 | 202.98 | 204.82 | **206.32** | 193.57 | 182.01 | 170.76 | 162.19 |

---

> > ### Author Response · Authors · 2024-08-13
> >
> > Dear Reviewer 6rnE,
> >
> > We would like to extend our appreciation for your time and valuable comments. We are eagerly looking forward to receiving your valuable feedback and comments on the points we addressed in the rebuttal. Ensuring that the rebuttal aligns with your suggestions is of utmost importance. Thank you for your dedication to the review process.
> >
> > Sincerely,
> >
> > Authors

---

> > ### Comment · Reviewer_6rnE · 2024-08-14
> > **Thanks!**
> >
> > Thank you for your point-by-point response, and the clarifications.
> > 1. Those details are very helpful, please add them to the figure in the final version!
> > 2. While the dimensions are indeed easy to specify, it is important to specify nonetheless for completion. It helps quickly clarify several confusion, such as which dimensions scalar variables are being added to, etc. Please add them to the final version.
> > 3. There was indeed some confusion about this because of the reasons previously stated. Your response is quite helpful, please add it to the final version. Since the same variable names such as F_theta are being specified for different architectures, it is critical to clarify such confusions in the main text.
> > 4. Good to know, thanks for the clarification.
> > 5. Thanks for the clarification.
> > 6. Good to know that \gamma was calculated empirically, would be helpful to add this maybe to the supplementary.
> > In light of the authors' response, I increase my score.

---

> > > ### Author Response · Authors · 2024-08-14
> > >
> > > We greatly appreciate you sparing time to read our rebuttal and give us an equally careful point-by-point response with an improved rating, and we will supplement these details to the final manuscript.

---

### Official Review · Reviewer_jEd7 · 2024-07-13

**Soundness:** 3
**Presentation:** 3
**Contribution:** 3
**Rating:** 5
**Confidence:** 2

**Summary:**

This work presents a framework for visual generative diffusion models, aiming at addressing the challenges associated with deep stacked networks in terms of numerical propagation errors and scalability.

**Strengths:**

1. Clear motivation

2. The authors provide a  theoretical analysis for their approach, including the use of continuous-time neural ODEs to demonstrate the relationship between residual-style network structures and generative denoising abilities.

3. Sufficient experiments: The paper is supported by extensive experimental evidence, which show that the proposed models achieve state-of-the-art performance on various generative tasks, including image and video generation.

**Weaknesses:**

1. While the introduction of gated residual parameters is innovative, it may also add complexity to the model, which could potentially make it harder to train or less intuitive for practitioners to understand.

2. How about the computation requirements for this scalability model?

3. Model parameters , GFLOPs should be mentioned.

**Questions:**

Listed in weakness.

---

> ### Author Rebuttal · Authors · 2024-08-06
>
> **Q1:** **Complexity of the model after introducing gated residual parameters**\
> **R1:** We understand the reviewer's concern, but that's actually unnecessary. Firstly, we want to highlight the introduction of these gated residual parameters is a _**simple yet meaningful**_ change to the common architecture of deep generative networks (e.g., _**U-Net**_ or _**Transformer**_), which only requires adding learnable scaling and bias tensors in each vanilla residual connection layer to fit the mean-variance scheduling dynamics of the proposed _**Denoising-Dynamics ODE**_. Secondly, we designed an error correction loss and integrated it into the conventional score matching loss at a smaller proportion (i.e., $\gamma=0.35$), balancing the model's denoising ability and error correction performance while ensuring that the model is easy to train.
>
> **Q2: Computation requirements**\
> **R2:** Neural-RDM focuses on helping existing or brand-new generative backbone networks support large-scale and deep-scalable training, which supports both full parameter training and partial parameter fine-tuning. In our experiments, _**2/4\*A100 80G GPUs**_ are respectively adopted to fine-tune the learnable gated parameters for image tasks (C2I & T2I) and video tasks (N2V & C2V) and _**8\*A100 80G GPUs**_ for the full parameter training from scratch.
>
> **Q3: Model parameters and GFLOPs**\
> **R3:** Thanks for this helpful suggestion, we have added the details of model parameters and GFLOPs in the table below, which will be included in the final version.
>
> | Task | Method | Parameters (M) | FLOPs (G) |
> | --- | --- | --- | --- |
> | Image Generation | Baseline (LDM-4) |  264.00  |  2,759.97   |
> |  | Neural-RDM-U (Ours) | 293.11 (**+11.02%**) | 2,931.91 (**+6.63%**) |
> |  | Baseline (Latte-XL/2) | 673.68 | 428.56 |
> |  | Neural-RDM-F (Ours) | 748.06 (**+11.04%**) | 457.26 (**+6.67%**) |
> | Video Generation | Baseline (Latte-XL/2) | 673.68 | 5,572.69 |
> |  | Neural-RDM (Ours) | 748.06 (**+11.04%**) | 5,603.80 (**+0.55%**) |

---

> > ### Comment · Reviewer_jEd7 · 2024-08-12
> >
> > Thanks for your reply. You have solved my concerns. It seems that the computational burden is not heavy.
> > I will keep my score.

---

> > > ### Author Response · Authors · 2024-08-12
> > >
> > > We sincerely thank you for sparing time and efforts reading our rebuttal and giving a response, thank you.

---

### Official Review · Reviewer_zq4Z · 2024-07-14

**Soundness:** 3
**Presentation:** 3
**Contribution:** 3
**Rating:** 6
**Confidence:** 4

**Summary:**

This paper addresses the problem of the numerical propagation errors of progressively deeper stacked neural networks for generative models. It proposes Neural Residual Diffusion Models (Neural-RDM), which introduced a series of learnable gated residual parameters to the common architectures of deep generative networks. Evaluation of the proposed method on image generation and video generation benchmarks demonstrate its superior performance over state-of-the-art methods.

**Strengths:**

- The proposed method is simple and well-motivated.
- The experiments on generation benchmarks are comprehensive, and the great results demonstrate the efficacy of the proposed method.
- The paper is clearly written.

**Weaknesses:**

- It is not clear what the specific definition of the "Scalability" columns in Table 1 and Table 2 is. Could you clarify the metrics represented by the cross, tick, and double tick symbols?
- Since the paper is targeting on addressing the scalability of deep generative models, it would be better if the scalability experiments could be enhanced. Figure 7 shows that the performance of the proposed method can be improved as the depth of the network increases. However, how does its scalability compare with baseline architectures? Also, does it show similar scalability on other tasks and datasets?
- As the authors have discussed in the limitation section, the numerical propagation errors caused by stacking network layers cannot be completely avoided. Therefore, it would be more accurate not to claim "enabling the networks to be *infinitely* stacked".

**Questions:**

See Weakness.

**Limitations:**

The authors have discussed clearly about the potential limitations and social impacts in A.4 and A.5.

---

> ### Author Rebuttal · Authors · 2024-08-06
>
> **Q1: Definitions and clarifications on scalability metrics** \
> **R1:**  Thanks for the helpful comment. We first want to clarify that the "Scalability" columns in Tab.1 & Tab.2 indicates the scaling capability (i.e., parameter scale and architecture stackability) of the evaluated models, meanwhile ensuring that errors do not accumulate as the network deepens, to facilitate the deep scalable training of the generative models (i.e., scaling law). Moreover, we want to re-clarify the meaning of the three metrics (i.e., _**cross**_, _**tick**_ and **_double tick_**): 1) GAN: Non-scalability (_**cross**_), limited by the instability of deep network training between the generator and the discriminator; 2) U/F-shaped: Low-scalability (_**tick**_), supporting a certain deep-scalable training, but error accumulation will occur as the network deepens during gradient update; 3) Ours: High-scalability (**_double tick_**), supporting deep-scalable training with the integration of error correction mechanism. We will make these points clearer in the final revision and hope the above clarifications can help understand our work better.
>
> **Q2: Supplements of scalability experiments**\
> **R2:** Thanks for this valuable suggestion, we have supplemented the scalability experiments versus the baseline (i.e., **_Latte-XL_**) in following table, benchmarking on more tasks (**_T2I_**, **_N2V_** & **_C2V_**) and more datasets (**_JourneyDB_**, **_Taichi-HD_** & **_UCF-101_**). As shown in the table, in the T2I task, as the network depth increases from 12 to 32, the IS-score of our Neural-RDM shows a significant improvement compared to the baseline, which implys the superiority of the error control mechanism in supporting deep scalable training. A similar trend can be observed in the N2V and C2V tasks, which consistently validate the deep scalability advantage of our Neural-RDM. We will supplement the above results to the final version to better illustrate the performance of the proposed method.
>
> | Depths | T2I in JourneyDB (IS$\uparrow$)  |  | N2V in Taichi-HD (FVD$\downarrow$) |  | C2V in UCF-101 (FVD$\downarrow$) |  |
> | --- | --- | --- | --- | --- | --- | --- |
> |  | Baseline | Neural-RDM (Ours) | Baseline | Neural-RDM (Ours) | Baseline | Neural-RDM (Ours) |
> | 12 | **103.18** | 94.20 | 236.83 | **191.04** | 923.38 | **896.33** |
> | 24 | 134.62 | **167.36** | 171.91 | **128.91** | 681.31 | **625.46** |
> | 28 | 195.03 | **206.32** | 159.60 | **91.22** | 477.95 | **461.01** |
> | 32 | 211.97 | **231.06** | 121.89 | **59.71** | 376.46 | **338.08** |
>
> **Q3: Regarding the imprecise claim.** \
> **R3:**  Thanks for this very thoughtful comment.  We will refine this claim to make it more precise in the final version.

---

> > ### Author Response · Authors · 2024-08-13
> >
> > Dear Reviewer zq4Z,
> >
> > We would like to extend our appreciation for your time and valuable comments. We are eagerly looking forward to receiving your valuable feedback and comments on the points we addressed in the rebuttal. Ensuring that the rebuttal aligns with your suggestions is of utmost importance.
> > Thank you for your dedication to the review process.
> >
> > Sincerely,
> >
> > Authors

---

### Author Rebuttal · Authors · 2024-08-06

We sincerely thank all the reviewers for sparing their time and efforts reading our paper and giving many insightful comments. We notice that all reviewers hold a positive view regarding the _**well motivation**_ and _**superior performance**_ of our model. The major questions are summarized below and addressed point by point, to help understand our work better.

---

### Decision · Program_Chairs · 2024-09-25

**Decision:**

Accept (poster)

**Comment:**

The paper addresses an essential problem of scaling up diffusion models, making use of gated residual mechanisms. The positive feedback of the reviewers, confirmed during the discussion period, indicates that the paper offers valuable insights and interesting method to achieve scaling of diffusion models. The authors are requested to include the results of the discussion of the reviewers in the final version of their paper, specifically related to the complexity of the method (empirical evaluation) as well as clarity of the presentation.